# The Role of Anti-PLA_2_R and Anti-THSD7A Antibodies in the Pathogenesis and Diagnostics of Primary Membranous Nephropathy: A Review of Current Knowledge for Clinical Practice

**DOI:** 10.3390/ijerph19095301

**Published:** 2022-04-27

**Authors:** Iwona Smarz-Widelska, Dariusz Chojęta, Małgorzata M. Kozioł

**Affiliations:** 1Department of Nephrology, Cardinal Stefan Wyszynski Provincial Hospital, 20-718 Lublin, Poland; i.widelska@interia.pl; 2Chair and Department of Medical Microbiology, Medical University of Lublin, 20-093 Lublin, Poland; dariusz.chojeta@gmail.com

**Keywords:** nephrotic syndrome, PMN, anti-phospholipase A2 receptor, anti-thrombospondin type I domain-containing 7A

## Abstract

Primary membranous nephropathy (PMN) is considered a major cause of nephrotic syndrome. The discovery of circulating autoantibodies directed against glomerular podocytes helped to classify them as autoimmune diseases. Over the past years, there has been an increasing significance of anti-Phospholipase A2 Receptor (anti-PLA_2_R), which has been detected in 70–80% of PMN cases, and relevance of anti-Thrombospondin type I domain-containing 7A (anti-THSD7A) even though they are present in 2–5% of patients. The results of clinical and experimental studies indicate that these antibodies are pathogenic. It radically changed the diagnostic and therapeutic approach. Measurement of antibody titers in the serum seems to be a valuable tool for identifying PMN and for the assessment of disease activity. By monitoring pathogenic antibodies levels rather than proteinuria or reduced glomerular filtration rate (GFR) as an indicator of glomerular disease, physicians would easier divide patients into those with active and inactive PMN disease and decide about their therapy. The aim of this review is to evaluate scientific evidence about the role of autoantibodies, namely anti-PLA_2_R and anti-THSD7A, as PMN biomarkers. The present manuscript focuses on PMN pathogenesis and key data of diagnosis, monitoring of the disease, and treatment strategies that are currently being used in clinical practice.

## 1. Introduction

Membranous nephropathy (MN) is a type of an organ-specific autoimmune disease of the glomeruli. It is associated with high levels of protein in the urine, in most cases in the nephrotic range (>3.5 g/d), and possible progression to end-stage renal disease in long-term follow-up [1,2]. In the remaining cases (70–80%), primary membranous nephropathy (PMN) is observed where the main role is played by PLA_2_R (phospholipase A2 receptor) and THSD7A (Human thrombospondin, type I, domain containing 7A) antigen. MN is the most common cause of nephrotic syndrome in worldwide prevalence in non-diabetic Caucasian adults (approximately 20–40% of cases), with the peak incidence at the age of 40–50 [3]. At the same time, it is very uncommon in the pediatric population, and children account for only 3% of patients with nephrotic syndrome [3,4,5,6,7]. In the other ~30% of cases, the disease is recognized as secondary MN and refers to cases involving other diseases (HBV-hepatitis B virus, SLE-systemic lupus erythematosus, sarcoidosis, cancer), exposure to drugs, or environmental pollution [3,4,8]. MN shares a common histopathological pattern characterized by immune complexes deposition in the subepithelial layer of the glomerular basement membrane. The turning point in the understanding of this disease was the discovery of autoantigens. The discovery began with *Beck* et al. in 2009, who demonstrated the presence of anti-PLA_2_R antibodies located on the surface of human podocytes, and then continued with the subsequent discovery in 2014 by *Tomas* et al., which demonstrated antibodies against thrombospondin type I, anti-THSD7A. It helped to develop new diagnostic methods, assessment of disease activity, and treatment outcomes [7,9,10]. To assist recognition of the disease, ideal diagnostic panel should include a screening of secondary MN cause, evaluation of the PLA_2_R/THSD7A antigen expression in renal tissue, and measuring serum anti-podocytes antibodies titer. Elevated anti-PLA_2_R levels can indicate the necessity for upgrading MN treatment for better long-term outcomes [11]. The aim of this review is a comprehensive analysis of the current literature concerning the role of autoantibodies (anti-PLA_2_R/anti-THSD7A) in the pathogenesis of membranous nephropathy and their use in the diagnostic and therapeutic process monitoring. The paper was divided into three main parts: pathogenesis of PMN (comparison of primary and secondary MN as well as the meaning of anti-PLA_2_R/anti-THSD7A), schedule of the diagnostic process (laboratory tools used), and how the course of disease and treatment efficacy can be controlled. This narrative review was prepared based on the scientific *PubMed* database and the use of *Google Scholar* as a support portal.

## 2. Pathogenesis

In recent years, research has significantly improved towards understanding of the pathogenesis of PMN [3]. Current concepts derived from earlier research were carried out 60 years ago, where experimental laboratory rat models with membranous nephropathy (HN, Heymann nephritis) were used, and which revealed IgG subepithelial deposits as part of immune complexes. In this rat model, megalin found on the surface of podocytes was identified as a target antigen for antibodies and responsible for the formation of glomerular subepithelial immune deposits [12]. These studies also demonstrated that complement plays a key role in the development of sub-lethal glomerular injury. Since megalin is not an antigen of human podocytes, there has been an ongoing search for target antigens of podocytes in humans. Evidence for this hypothesis was provided by the studies of *Beck* et al., who found that the phospholipase A2 type M receptor was the main target antigen of podocytes in patients with PMN [7]. It has been shown that anti-PLA_2_R are mainly of IgG4 subclass, simultaneously present both in the circulation and in glomerular deposits. Furthermore, glomerular IgG4 are generally considered as being unable to activate complement via the classical route [13]. This finding suggested complement activation is going via two other pathways: lectin or an alternative pathway [14]. The lectin pathway is essential due to the detection of MBL (mannose-binding lectin) and C4b, a mannose-binding lectin activation product in the renal glomeruli [15,16]. It was observed by *Hayashi* et al., who identified the presence of MBL deposits in the glomeruli at the same location as PLA_2_R [16]. Due to the unknown pathomechanism leading to complement activation and glomerular damage, *Haddad* et al. created an in vitro model based on human PLA_2_R-expressing podocytes and using anti-PLA_2_R IgG4 positive serum from PMN patients. It was demonstrated that improperly glycosylated IgG4 promotes activation of the complement lectin pathway and induces glomerular damage [17]. Disorders that affect glycosylation of human IgG have been so far demonstrated in other autoimmune diseases, e.g., rheumatoid arthritis [18]. Perm-selectivity of the glomerular filtration barrier is provided by foot processes of podocytes, which are organized by the actin cytoskeleton and connected by a slit diaphragm. Glomerular diseases with proteinuria, including membranous nephropathy, are characterized by podocyte foot processes effacement. It is a result of altered transmembrane proteins-nephrin, podocin, and NEPH1 and proteins regulating the cytoskeleton dynamics, such as synaptopodin, L-actinin 4, and dynamin. The data of *Haddad* et al. were relevant for understanding the role of anti-PLA_2_R IgG4 in complex formation, which attacks the podocyte membrane (MAC-membrane attack complex; C5b-9) [17]. Proteolytic degradation of podocyte proteins, including synaptopodin [19], nephrin [20], NEPH1 [17], CD2AP [21], and dynamin [22], are induced, which alters the actin cytoskeleton dynamics and disrupts slit diaphragm. This consequently leads to glomerular damage [17,23].

### 2.1. Primary and Secondary Membranous Nephropathy

Membranous nephropathy is a heterogeneous disease that manifests itself in the absence of an established cause (primary MN) or the course of some clinical conditions (secondary MN), such as infections (e.g., HBV, HCV—hepatitis C virus), autoimmune diseases (systemic lupus erythematosus, Hashimoto’s thyroiditis, Sjogren’s syndrome), cancers, or drug intoxication [12]. Other factors such as air pollution, cigarette smoking, or frequent respiratory infections may also contribute to the development of MN. The increased exposure of the immune system to contact with foreign antigens may facilitate the occurrence of the “*antigenic mimicry*” phenomenon [24]. The role of this phenomenon in the pathogenesis of MN is even more likely to be considered if we look at podocyte antigens PLA_2_R and THSD7A attending in PMN, which share a common epitope at the N-terminus [25]. The discovery of target antigens has created a greater heterogeneity in membranous nephropathy that can be defined by detecting autoantibodies using serological methods or by renal expression of PLA2R and THSD7A in renal biopsy specimens [2,26]. Patients with the secondary form of MN have not presented positive blood test results for anti-PLA2R in the overwhelming majority of cases. Nevertheless, according to a few scientific reports, such a positive antibody titer was noted [27,28]. Therefore, seropositivity should not completely exclude the search for secondary causes, especially in patients over 60 years of age and with risk factors for developing neoplastic disease [27]. Morphological alterations of the glomerulus are also important in the differential diagnosis. The predominance of IgG4 deposits in immunofluorescence microscopy is generally more common in primary nephropathy, while IgG1-IgG3 deposits are more often detectable in secondary nephropathy [29]. According to the mechanism of the lectin pathway in the deposits in the renal glomeruli, PMN also finds components C3, C4d, and C5b-C9 (membrane attack complex), but there should not be complement components C1q [4,5,16]. Nevertheless, there have been reported PMN cases with deposits composed of IgG1 immunoglobulins and C1q components [30]. The explanation for this phenomenon is the hypothesis that at the beginning of the disease development, the classical route and IgG1 antibodies play a greater role than the lectin route and IgG4 antibodies [6]. The statement is in line with the assessment that in the course of the autoimmune process, the dominant subclass of antibodies evolves in the following order: IgG3 → IgG1 → IgG2 → IgG4 [24]. *Ryan* et al. showed that in the early stages of PMN (stage I), the IgG1 subclass is dominant and is usually associated with a lack of PLA_2_R, contrary to later stages (stages II–IV), where IgG4 and positive PLA_2_R are dominant. The authors suggest that low levels of PLA2R trigger an IgG1 response that may later switch to an IgG4 response during disease progression [31]. A summary of the above-described features of the primary and secondary forms of MN is included in Table 1.

### 2.2. Anti-THSD7A Antibodies

Thrombospondin type-I domain-containing 7A is a 250 kDa multidomain protein expressed on podocytes. The localization of the THSD7A transmembrane protein in podocytes is foot processes and its function is to increase the mutual adhesion of cells. Therefore, anti-THSD7A antibodies are likely to lead to structural and functional changes in the permeability of slit diaphragm to plasma proteins [35]. The analysis of data from the literature shows a relationship between the THSD7A antigens and their antibodies with neoplasm, while this link has been weak with PLA2R antigen and anti-PLA2R antibodies [36,37]. Scientific reports have linked secondary MN associated with the presence of anti-THSD7A with proliferative processes such as rectal cancer, gallbladder cancer, or with ALHE-angiolymphoid hyperplasia with eosinophilia [37,38,39,40]. Concerning the latter, the role of vascular endothelial growth factor (VEGF-A) and its neovascularization is likely in cases of MN and ALHE coexistence [39]. The association between neoplasm and MN is probably the result of the expression of THSD7A antigens within the tissues of some tumors [41]. On basis of the auto-aggressive immune process, the production of circulating autoantibodies in the blood may then occur, but it is not clear whether this primary immunization occurs within the tumor tissue or the kidney tissue [37]. Nevertheless, it should be considered that exact pathogenesis and the causal link of the coexistence of these two diseases remain unclear at the moment. There is no doubt that the discoveries described above should encourage the initiation of diagnostics towards secondary MN or the coincidence of PMN and the neoplastic process [42]. All of this gives MN related to THSD7A a character of a specific type of paraneoplastic syndrome or, more rarely, make it a revelator of neoplastic disease. Everything depends on the timing of the onset of kidney disease and the diagnosis of neoplasm. This should prompt an active search for the hidden neoplastic process in this group of patients, especially when the patient has circulating anti-THSD7A antibodies in the blood [43]. An example is a Japanese analysis where eight patients with MN related to anti-THSD7A developed a tumor within three months of the study. An important observation was also the predominance of women in the MN group associated with THSD7A compared to the group with MN- and anti-PLA2R-positive results, which was dominated by men [44].

### 2.3. Anti-PLA_2_R Antibodies

PLA2R is a 180-kDa transmembrane glycoprotein belonging to the mannose receptor (MR) family, and it is expressed on human podocytes. Autoantibodies produced against PLA_2_R proved to be a promising biomarker for the diagnosis and for monitoring MN activity. The MN related to PLA_2_R defines patients with increased levels of anti-PLA_2_R antibodies circulating in the blood as well as those with the presence of the PLA_2_R antigen in the kidney biopsy specimen treated by special staining. The group of patients with the presence of autoantigen but no autoantibodies accounts for 10–15% of PMN cases. It is thought that in an early stage of the disease, when immunization and antibody formation has not yet occurred, it is important to perform serial laboratory determinations anti-PLA_2_R for better control [3,5]. Moreover, lab testing is necessary as well in patients with viral hepatitis-MN. Anti-PLA_2_R antibodies are not that unusual in it, howeverthe findings of *Berchtold* et al. suggest that infection of HBV might be the trigger of this autoantibody [45]. Scientists suggested that PLA_2_R and THSD7A are not mutually exclusive. *Larsen* et al., while working on a group of native renal biopsies with MN, showed dual positive staining for these two antigens although only in 1% of cases. At the same time, there was a full correlation between positive staining results and corresponding autoantibodies circulating in the blood [46]. In addition, *Wang* and colleagues, in a cohort study including 578 patients with MN, observed two patients (0.3%) having anti-PLA_2_R and anti-THSD7A at the same time. This study showed no immunological cross-reactivity between these antibodies against THSD7A and PLA_2_R, which suggests the emergence of a dual immune response and the importance of the specific antigen during immune subtyping [47]. Similarities and differences between anti-PLA_2_R and anti-THSD7A are presented in Table 2.

## 3. Diagnosis

The kidney biopsy remains the gold standard for diagnosis MN even for the patients positive to anti-PLA_2_R/THSD7A. In individuals who are seronegative, biopsy gives a benefit of fully checking autoimmune background of the disease by staining and visualization of the presence of THSD7A and PLA_2_R antigens [5]. Serological testing, in comparison to invasive diagnostic methods is a relatively simple, repeatable, and quite fast tool, so it allows for a relatively effective diagnostic–therapeutic process in patients with PMN. The determination of autoantibody will be even more important when we consider cases where biopsy is contraindicated or not recommended, e.g., one kidney, the patient’s clinical poor condition, or some severe complications of a nephrotic syndrome, such as pulmonary embolism [4]. From currently available diagnostic tests, which measure antibody titers in serum, the enzyme-linked immunosorbent assay (ELISA) provides a quantitative result and gives an advantage over semi-quantitative evaluation by indirect immunofluorescence test (IIFT) or Western blot technique [34,49,50]. Determination methods for anti-PLA_2_R are as follows: sensitivity in the range of approximately 60–80% and specificity in the range of 90–100% (the basis for calculations was a kidney biopsy, which results in the determined diagnosis of PMN) [4,50,51,52,53,54]. The diagnostic value of anti-PLA_2_R tests is currently much better estimated and developed than for anti-THSD7A. In determining anti-PLA_2_R, both of the above-mentioned types of tests are available on the market (commercial), while the tests for determining anti-THSD7A have scientific status [34]. It should be noted that detection range and values in anti-PLA2R ELISA test are subject to discussion. Many researchers indicate that adopting values lower than those recommended by test manufacturers would have better diagnostic value (increased sensitivity while maintaining specificity at a comparably high level) [53,54,55]. The studies reveal a prominent role of serological testing by demonstrating anti-PLA2R titer correlation with a clinical disease manifestation, laboratory parameters, and kidney biopsy results in patients [49]. It is worth mentioning that all of the evidence is much better documented for anti-PLA_2_R than for anti-THSD7A. When antibodies for PLA_2_R were present, not only the positive result itself but the level of them provides prognostic information about the severity of the disease, and it can serve as a biomarker of treatment effectiveness [56]. A higher sera concentration supports a worse clinical course, a weaker response to immunosuppressive therapy, and a longer time to remission [57]. It is also associated with a high risk of progression of kidney failure [39,58]. For this reason, the initial high and increasing antibodies titer act as an indicator for starting immunosuppressive therapy as fast as possible because there is a low chance for spontaneous remission and a higher risk of losing kidney function [33,34,59]. During clinical remission, relapse rates also correspond with levels of circulating antibodies. Patients who become negative after treatment have a lower relapse rate [60,61].

## 4. Monitoring the Course of Disease and Treatment Efficacy

PMN is a chronic disease that proceeds in the form of alternating remissions (spontaneous or induced by immunosuppressive treatment) and relapses. The discovery of anti-PLA_2_R/THSD7A autoantibodies was a big step forward in the diagnosis and treatment of patients because of the strong correlation of the antibody titer with the disease activity and pharmacotherapy effects. This applies to both anti-PLA2R [33,49,51,62,63,64,65,66,67,68] and anti-THSD7A [34,47]. An advantage of serological testing is the fact that changes in the autoantibodies levels precede any changes in biochemical parameters, in particular changes in the degree of proteinuria [33]. The decreasing concentration of anti-PLA2R is indicative of immune remission and overtakes clinical remission in several weeks/months. On the other hand, in cases where remission has been achieved, the reappearance of anti-PLA_2_R antibodies is a predictor of clinical relapse [33,62,64]. This shift and the postponed interplay between immune and clinical remission indicates the prolonged time required for the formation of the immune deposits, which will induce proteinuria. Conversely, sufficient time must elapse to remove subepithelial deposits, restore per-selective function of the glomerular filtration membrane, and resolve proteinuria. Therefore, proteinuria is a poor clinical biomarker for monitoring disease activity and treatment efficacy [69]. Monitoring of the disease activity should be based on estimated GFR, proteinuria, and as well values of anti-PLA_2_R. Risk stratification is very essential for clinical decisions. According to the data, we can distinguish some categories of the disease progression: low risk, moderate risk, and high risk (Table 3).

The effectiveness of immunosuppressive therapy based on the antibodies level can be initially assessed as early as 1–2 months after the treatment initiation. The indicator of the response to treatment is a decrease in the antibody titer accompanied by albumin increase in blood and proteinuria decrease [66,67]. The latest discoveries have therapeutic implications. Immunosuppressive therapies, including alternating corticosteroids with alkylating agents and calcineurin inhibitors (CNI), are effective in reducing proteinuria, but their use may result in numerous side-effects and a high relapse rate. New B-cell-targeted therapies are increasing in importance nowadays. Current treatment options include rituximab. It is a chimeric CD20 mouse/human monoclonal antibody (MAb) against CD20 receptor on the surface of B cells. The medication is relatively effective in achieving remission, reducing relapse rate, and maintaining renal function with a favorable safety profile. The influence of treatment has so far been demonstrated in many multi-center studies [71,72]. Monitoring anti-PLA2R sera concentration seems to be a good and sensitive biomarker that predicts response to rituximab treatment. In the case of B-cell resistance to rituximab therapy, other treatment options should be considered [73]. One of them can be the use of a selective inhibitor of proteasome (bortezomid). This novel treatment option could help to reduce proteinuria if MN would recur after renal transplantation [74]. Currently, research focuses on the effectiveness of anti-CD20 antibodies of the second and third generation (obinutuzumab, ofatumumab) [73]. Owing to serological testing, it is possible to individualize the treatment of patients with PLA_2_R-associated PMN while reducing unnecessary exposure to immunosuppressive therapy. A slight reduction of the anti-PLA_2_R titer (<50%) suggests the need to change immunosuppressive treatment, and rapid decrease (>90%) in less than 6 months is the basis for the decision to terminate the immunosuppressive treatment early [33]. There are new PMN therapies on the horizon. The blockade of the lectin pathway may be a breakthrough in treatment both as a substitute and in combination with existing immunosuppressive drugs.

### Kidney Transplantation

Membranous nephropathy (MN) is one of the many kidney diseases that can lead to failure. A group of patients developing end-stage renal disease indicate renal replacement therapy such as kidney transplantation. Relapse of the underlying disease in an organ transplant is relatively common; e.g., focal segmental glomerulosclerosis (FSGS) can develop. MN in transplant recipients may develop *de novo*, which is uncommon (1–2%), or it may recurrent, and the frequency of this phenomenon is estimated at 40–50% of the transplanted organs [75,76,77,78]. The presence of anti-PLA_2_R is observed in most cases of recurrence MN, while they are negative in *de novo* MN. Immunological factors such as a viral infection or renal ischemia-reperfusion injury may likely be a trigger and uncover hidden podocyte antigens for the immune system of the recipient (auto-or alloantibodies against podocytes) [78,79]. The risk of PMN recurrence in the received kidney is even greater when autoantibody titers anti-PLA_2_R/THSD7A are high in the peri-transplant period or produced after transplantation [80]. Findings of *Quintana* et al. confirmed PMN recurrence in patients with high anti-PLA_2_R concentration (cut-off-45 U/mL) in the pre-transplantation period [77], and Grupper et al. also observed it, but at the same time, there was no link noted between the progression of recurrent disease and antibodies [75]. It is estimated that one-third of patients with recurrent MN have no progression, and there is no need for supplementary immunosuppression, but for the rest, there exists a high risk of graft loss [33]. It is considered that the titer of circulating anti-PLA_2_R has a meaning at the transplantation period. Antibodies, by reaching receptors on podocytes of the allograft, form faster subepithelial deposits. *Blosser* et al., in their case report, indicated quick recurrence of PMN related to anti-PLA_2_R after transplant. Preexisting autoantibodies influenced renal injury even before clinical evidence [81]. Posttransplant immunosuppression has an impact on disease activity, but the mechanism is not fully explained. Determining the serum concentration of antibody of transplant patients can be a useful tool to make a faster decision to introduce immunosuppressive therapy at an early stage of the disease development [33,81]. Treatment of choice for suspected recurrent PMN in the transplanted kidney is anti-CD20/rituximab [75,80], which gave good results also as a form of therapy in the general population of patients [82]. Anti-CD20 monoclonal antibodies influence B-cell depletion and can be used in native kidneys membranous nephropathy treatment and as well in renal allografts recipients [75]. Using it in recurrent MN seems to be promising. According to studies by *Debiec* et al., it helped clinical improvement and decreased anti-PLA_2_R levels. However, scientists also described a case of anti-PLA_2_R1 disappearance while the patient underwent conventional immunosuppressive treatment without rituximab despite a high antibody titer at the time of transplantation [83].

## 5. Summary

Primary membranous nephropathy (PMN) is recognized as one of the leading causes of nephrotic syndrome in adults. Advancements in understanding the mechanisms of PMN in the past decade have resulted in a new diagnostic and therapeutic approach. The role of the described anti-PLA_2_R and anti-THSD7 antibodies is gaining importance, as confirmed by numerous publications in which studies and meta-analyses demonstrate the autoantibodies can be initially assigned as a highly specific biomarker of this disease. Nevertheless, there are several limitations of their use because not all of the patients with MN have elevated antibody titers. Antibodies’ presence indicates primary MN but does not exclude secondary MN. Despite many scientific achievements, kidney biopsy is still the gold standard for the diagnosis of PMN. While anti-PLA_2_R is increasingly included in clinical practice as an additional assay, there is a need for further research dedicated to anti-THSD7A. In the presented paper, many relevant studies were reviewed but not all of them. Despite this weakness, it can be concluded that serological diagnostic has become an important and indispensable tool for monitoring the course of disease and treatment process and opening the window towards personalized medicine. Identification of other target antigens in PMN continues to evolve, allowing for better diagnosis and new treatment strategies for this disease.

## Figures and Tables

**Table 1 ijerph-19-05301-t001:** Comparison between the primary and secondary forms of membranous nephropathy [3,4,5,6,32,33,34,35,36].

Characteristic	Primary MN	Secondary MN
rate of total MN cases	70–80%	20–30%
cause of disease	autoimmune disease	underlying diseases and drug or poison exposure
directed treatment	immunosuppressive therapy	treatment focused on the underlying disease
dominant antibody subclass detected in deposits of immune complexes	IgG4	IgG1, IgG2, IgG3
detectability of C1q in deposits of immune complexes	very rare	present
the role of the classical pathway of complement activation in disease pathogenesis	less probable	more probable
hypotheses about the formation of immune complexes	penetration of autoantibodies against podocyte through the glomerular membrane	○ decay and depositionof immune complexes circulating in the blood ○ a build-up of foreign antigens in the sub-epithelial layer of the glomerulus and penetration of antibodies against them

**Table 2 ijerph-19-05301-t002:** A comparison of anti-PLA_2_R and anti-THSD7A [10,34,36,37,38,39,40,41,43,47,48].

Characteristic	Anti-PLA_2_R	Anti-THSD7A
*podocyte autoantigen*	M-type phospholipase A2 receptor (PLA_2_R)—185 kDa	thrombospondin, type I, domain containing 7A (THSD7A)—250 kDa
frequency detection in patients with PMN	~70%	~2.5–5%
year of discovery	2009	2014
role of serology in the diagnosis	high role (better described)	lower role (less described)
link to neoplastic processes	low	strong
laboratory methods	IIFT, ELISA, Western blot	IIFT, ELISA, Western blot
dominant antibodies subclass	IgG4	IgG4

**Table 3 ijerph-19-05301-t003:** Risk evaluation of disease progression [3,70].

Risk Category	eGFR	Proteinuria	Anti-PLA_2_R Titer *
low	normal/stable	<3.5 g/day	serial measurements persistently low
moderate	normal/stable	>3.5 g/day and no reduction (>50%) after 6 months of antiproteinuric therapy	medium levels of serial measuarements
high	<60 mL/min/L, 73 m^2^	>8 g/day	high levels of serial measuarements
	>60 mL/min/L, 73 m^2^	>3.5 g/day and serum albumin < 25 g/L	
very high	rapid deterioration of kidney failure	nephrotic syndrome	

* Cutoff values are not validated. Antibodies concentration should be measured at 3–6-month phases.

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
