# Peer review of "The Role of Anti-PLA2R and Anti-THSD7A Antibodies in the Pathogenesis and Diagnostics of Primary Membranous Nephropathy: A Review of Current Knowledge for Clinical Practice"

_ijerph, 2022, doi:10.3390/ijerph19095301_

Round 1
Reviewer 1 Report
Overall comments to the authors:
In this article, Smarz-Videlska I et al review the current knowledge about anti-PLA2R and anti THSD7A antibodies. The article has clinical relevance since it gives an updated and practical assessment of the field and it is also well-structured.
On the other hand, some changes are needed in order to expand some relevant points.
Major comments:
- I would suggest the authors to expand the kidney transplant section (4.1).
- a) Some authors have described an anti-PLA2R cut-off level during the pre-transplant period to accurately predicte PMN recurrence after transplantation. The authors should explain the possible role of anti-PLA2R level at the moment of kidney transplant (either with living or cadaveric donor) in the choice of rituximab induction
- b) The role of antiPLA2R in the differential diagnosis between secondary and primary membranous nephropathy in kidney transplant recipients biopsies in patients with unknown etiology of their native chornic kidney disease should be discussed, since transplant recipients are at an increased risk of cancer.
- Authors should include a new table that include the risk of progression to chronic kidney disease based on antiPLA2R level, proteinuria and glomerular filtration rate.
Minor comments:
-Please change the acronym PNB for PMN. This spelling mistake can be found in page 1-line 25 and in page 3-line 125.
Author Response
The authors sincerely thank the Reviewer for all comments and critical assessments of our paper.
We have done our best to improve it. Below are our responses to each concern.
Please see the attachment.

Reviewer 2 Report
This is a comprehensive review summarizing the predictive value of anti-phospholipase A2 receptor (anti-PLA2R) and anti-thrombospondin type I domain-containing 7A (anti-TSHD7A) antibodies with regard to incidence of primary membranous nephropathy (PMN), which is an important cause of nephrotic syndrome. Authors utilized the PubMed database to evaluate the correlation of these antibodies with disease activity and pathogenesis in addition to their diagnostic role. The manuscript is well-documented with relevant subheadings and a complete bibliography at the end. However, there are many grammatical errors throughout the manuscript that undermine the credibility of the paper. I have the following suggestions, which I believe might contribute to the improvement of the paper:
1) Please define the following abbreviations when those appear first in the text: PNB (line 25), HBV (line 42), SLE (line 42), MBL (line 80), MAC (line 96), HCV (line 104)
2) The following sentences do not sound grammatically correct, please re-write:
- … and the main role play … (line 35)
- Larsn et al. on a group of … (line 187)
- As well Wang et al. in a cohort … (line 190)
- One of the drugs … (line 263)
3) I can suggest correcting and enriching the following sentence (line 96-99) with the appropriate citations:
- Proteolytic degradation of … >> Proteolytic degradation of podocyte proteins including synaptopodin [Faul C, et al., Nat Med. 14:931-8, 2008], nephrin [Tan RJ, et al. JCI Insight 4:e122399, 2019], NEPH1 [Haddad G, et al. J Clin Invest 131:e140453, 2021], CD2AP [Yaddanapudi S, et al., J Clin Invest 121:3965-80, 2011] and dynamin [Sever S, et al. J Clin Invest. 117:2095-104, 2007] are induced, which alters the actin cytoskeleton dynamics and disrupts slit diaphragm. This consequently leads to glomerular damage [Yu SM, et al. Front Med (Lausanne) 5:221, 2018].
4) Please make the following corrections in the related sentences:
- strategies use >> strategies that are currently being used (line 26)
- demonstrate >> demonstrated (line 48)
- sera >> serum (line 52)
- main three parts >> three main parts (line 57)
- tools which are used >> tools used (line 60)
- improved understanding >> improved towards understanding (line 64)
- as unable >> as being unable (line 77)
- an alternative or lectin >> lectin or an alternative (line 79)
- diseases, like e.g. >> diseases, e.g., (line 88)
- foot processes podocytes >> foot processes of podocytes (line 89)
- the last one >> the latter (line 153)
- to initiation >> the initiation (line 161)
- What is more, >> Moreover, (line 183)
- result in >> results in (line 217)
- gaining in importance >> gaining in importance (line 263)
- B cells surface >> surface of B cells (line 265)
- would recurrent >> would recur (line 272)
- research studies are performed on >> research focuses on (line 273)
- FSGS can show up >> FSGS can develop (line 286)
- or their reappearance after transplantation >> or produced after transplantation (line 290)
- antibody sera concentration >> serum concentration of antibody (line 290)
5) Please correct the following typos:
- TSHD7A >> THSD7A (line 16)
- PLAR2R >> PLA2R (line 24)
- Perselectivity >> Permselectivity (line 89)
- NB >> MN (line 107)
- glomerular >> glomerulus (line 120)
- kDA >> kDa (line 143)
- PBN >> PMN (line 162)
- anty >> anti (line 183)
- laboartory >> laboratory (Table 2)
- Owning >> Owing (line 274)
Author Response

(The authors gave the same response as above.)
